# Label-Free LSPR-Vertical Microcavity Biosensor for On-Site SARS-CoV-2 Detection

**DOI:** 10.3390/bios12030151

**Published:** 2022-02-28

**Authors:** Yuqiao Zheng, Sumin Bian, Jiacheng Sun, Liaoyong Wen, Guoguang Rong, Mohamad Sawan

**Affiliations:** 1CenBRAIN Lab, School of Engineering, Westlake University, Hangzhou 310024, China; zhengyuqiao@westlake.edu.cn (Y.Z.); biansumin@westlake.edu.cn (S.B.); 2School of Engineering, Westlake University, Hangzhou 310024, China; sunjiacheng@westlake.edu.cn (J.S.); wenliaoyong@westlake.edu.cn (L.W.)

**Keywords:** biosensor, COVID-19, SARS-CoV-2, vertical microcavity, localized surface plasmon resonance, nanoporous gold, artificial saliva

## Abstract

Cost-effective, rapid, and sensitive detection of SARS-CoV-2, in high-throughput, is crucial in controlling the COVID-19 epidemic. In this study, we proposed a vertical microcavity and localized surface plasmon resonance hybrid biosensor for SARS-CoV-2 detection in artificial saliva and assessed its efficacy. The proposed biosensor monitors the valley shifts in the reflectance spectrum, as induced by changes in the refractive index within the proximity of the sensor surface. A low-cost and fast method was developed to form nanoporous gold (NPG) with different surface morphologies on the vertical microcavity wafer, followed by immobilization with the SARS-CoV-2 antibody for capturing the virus. Modeling and simulation were conducted to optimize the microcavity structure and the NPG parameters. Simulation results revealed that NPG-deposited sensors performed better in resonance quality and in sensitivity compared to gold-deposited and pure microcavity sensors. The experiment confirmed the effect of NPG surface morphology on the biosensor sensitivity as demonstrated by simulation. Pre-clinical validation revealed that 40% porosity led to the highest sensitivity for SARS-CoV-2 pseudovirus at 319 copies/mL in artificial saliva. The proposed automatic biosensing system delivered the results of 100 samples within 30 min, demonstrating its potential for on-site coronavirus detection with sufficient sensitivity.

## 1. Introduction

Periodic outbreaks of the human coronavirus have attracted global attention. The outbreak of severe acute respiratory syndrome coronavirus 2 (SARS-CoV-2) in December 2019 has led to a severe global crisis. Biosensors with high detection sensitivity, throughput, and processing speed are urgently needed for onsite detection of SARS-CoV-2 in public places to deal with pandemics [1]. Currently available methods for SARS-CoV-2 detection include electrical [2], electrochemical [3,4,5], and optical [6] approaches. For example, a graphene-based biosensor detected SARS-CoV-2 in turbinate swab samples from infected patients in a low viral concentration (242 copies/mL) within seconds [7], while an electrochemical biosensor detected SARS-CoV-2 gene at levels as low as 200 copies/mL [3]. Electrical and electrochemical biosensors are sensitive, disposable, and inexpensive for public heath monitoring, although the latter may take longer time to complete a measurement [3,4,5]. When it comes to on-site detection, optical biosensors are more promising, on the one hand, in being sensitive and rapid, and on the other hand, in holding potential of achieving “high-throughput”, due to full automation. Moreover, optical biosensors can have reliable biosensing performance due to their negligible effects from electromagnetic interference.

Optical signals include colorimetry, absorption, reflectance, chemiluminescence, and fluorescence, and can be label-free or label-based [8]. Among them, optical plasmonic biosensors based on localized surface plasmon resonance (LSPR) detect target analytes by measuring the optical responses in the vicinity of metal nanoparticles. The response signal is correlated with the change of the refractive index (RI) in a local solution [9], or the interaction of light with attached molecules [10]. The RI change is instantaneous and can be tracked via the plasmonic peak shift on spectrum, indicating the potential to deliver quick results [11]. In a recent study, a LSPR biosensor was reported to achieve a high sensitivity with a limit of detection (LoD) of 0.047 μg/mL within 30 min for detection of dengue antigen. Another type of RI-sensing optical sensor based on the vertical resonance cavity has also been reported, where the vertical cavity serves as the transducer. The compact structure of vertical cavity-based sensors allows them to be designed in a small size, miniaturizing the biosensing system to the most extent [12].

Enhancing the sensitivity of optical biosensors is of particular importance for SARS-CoV-2 detection. In addition to adopting high-affinity bioreceptors, proper blocking, and signal amplification strategies, appropriate design of the sensor structure with optimized parameters is crucial [13]. The performance of a plasmonic-based biosensor largely depends on the surface morphology or the nanostructure design, as a higher amount of plasmons excites a stronger optical response [14]. Moznuzzaman et al. found that the sensitivity of surface plasmon resonance-based biosensors can be optimized by adjusting the silver thin film thickness [15]. Chowdhury et al. have reported that the Au nanoparticle size and interparticle distance have a strong influence on fluorescence enhancement [16]. The virus detection performance of LSPR-based biosensor was carefully reviewed in a recent publication by Takemura [14]. Generally, a controllable metal nanoparticle size and interparticle distance are desirable for LSPR-based biosensors. Introducing a hybrid resonance mode can enhance the detection signal by limiting surface loss as well [17]. Buzavaite-Verteliene et al. immobilized an Au thin film on a distributed Bragg reflector (DBR) surface. The design generated Tamm plasmon–polariton resonance at the photonic crystal and metal layer interface, and greatly reduced the energy loss caused by the absorption and scattering of the Au thin film [18]. Nanoporous gold (NPG) has been used as a high-performance electrode due to its high surface-to-volume ratio [19], and has been applied to excite surface plasmon resonance to increase the sensitivity of biosensors [20]. However, biosensors based on surface plasmon polariton still rely on couplers to be excited, in addition to the need for a temperature controller during the process [14]. At the same time, a compact LSPR biosensor structure with low sample amount required is promising in biosensing system miniaturization.

In this study, we designed a vertical microcavity-LSPR hybrid biosensor for onsite SARS-CoV-2 detection within 30 min. To achieve affordability, we adopted a low-cost technique to form NPG with various surface morphologies on SiO_2_ vertical microcavity (Figure 1A), as elaborated in detail in the following Materials and Methods section. The method in this work can form NPG with controllable surface morphologies with high speed. The diagram of the proposed biosensor is shown in Figure 1B. Apart from serving as a RI sensor and miniaturized biosensor substrate, the microcavity also generates high-quality resonance mode, which can reduce the effect of approximation errors and electrical noise on the plasmon peak shift calculation. The plasmon phenomenon of NPG significantly enhances the local electric field and the resonance quality. In the present study, NPG thin films with different porosities were coated on a microcavity wafer. We used simulations to study the RI sensing performance of different morphology of NPG deposited biosensors. In the real experiment, SARS-CoV-2 pseudovirus spiked in artificial saliva were used to further investigate the relationship between biosensor surface morphology and optical signal response, and to pre-clinically validate the designed hybrid biosensor. 

## 2. Materials and Methods

In this section, we introduce the materials, reagents, and facilities used in this work, as well as the hybrid biosensor fabrication and the biosensing system setup.

The 11-mercaptoundecanoic acid, 2-(N-morpholino) ethane sulfonic acid (MES), and 1-(3-dimethylaminopropyl)-3-ethylcarbodiimide hydrochloride (EDC) reagents were purchased from Macklin. N-hydroxy succinimide (NHS) and bovine serum albumin (BSA) were bought from Sigma. The capture monoclonal antibody (40150-D006), SARS-CoV-2 pseudovirus (PSV-001), and MERS-CoV spike protein (40071-V08B1) were purchased from Sino Biological (Beijing, China). The artificial saliva (A7990) was obtained from Solarbio Life Science (Beijing, China). The physical vapor deposition (PVD) (ZD-400 single chamber high vacuum resistive evaporator) was from Shenyang Kecheng Vacuum Tech Co. Ltd (Shenyang, China). The halogen light source, spectrometer (PG2000-Pro), and Y-shaped optic fibers were bought from Idea Optics (Shanghai, China). The robotic arm (Z-Arm 1632) was purchased from HITBOT (Shenzhen, China). The blade dicing machine (DS616) was bought from Shenyang Heyan Technology Co., Ltd. (Shenyang, China).

### 2.1. Vertical Microcavity-LSPR Biosensor Platform

The process flow in fabricating microcavity-LSPR biosensor is shown in detail in Figure 1A. In brief, four main steps were adopted, including electrochemical anodization, oxidation, deposition, and biofunctionalization. The construction steps were performed as published previously [6], with some modifications, described as follows.

#### 2.1.1. NPG Formation

After electrochemical anodization and oxidation, we deposited NPG thin film on the microcavity. NPGs have been reported to enlarge the effective electrode–electrolyte contact area due to their high surface-to-volume ratio [21]. In the present study, the deposition was done by PVD, which has higher deposition rates than sputtering and can form more consistent films in between batches compared with electroplating. The process steps in fabricating NPG coated biosensor are shown in Figure 1A. The adhesive layer, barrier layer, and Au–Ag alloy thin film were deposited in sequence. The adhesive layer provides good adhesion between the microcavity and the deposited layer. After the deposition of the barrier layer, we simultaneously deposited Ag and Au in different ratios. Then, Ag was selectively dissolved by merging the wafer surface in 69% HNO_3_ for 30 min. Finally, the NPG-coated biosensor die was cut by the blade dicing machine into 8 × 8 mm pieces.

#### 2.1.2. Surface Functionalization

For functionalization, we adopted the method we proposed before with minor modifications [22]. Biofunctionalization was performed through a four-step process: carboxylic group functionalization with 11-mercaptoundecanoic acid; carboxylic group activation using a 0.4 M EDC/0.1 M NHS mixture solution; antibody immobilization; blocking. In particular, 1 µg/mL of SARS-CoV-2 antibody was prepared in the buffer containing 0.2 M NaHCO3 and 0.1 M NaCl, pH 8.0. The biosensor die after NPG deposition was immersed in the diluted antibody solution for 30 min at 25 ℃.

### 2.2. Numerical Modeling of the Microcavity and NPG Parameters

The antigen-antibody binding event was characterized by the change of RI, which was obtained by calculating the average shift of six characteristic valleys (the calculation is introduced in Section 2.4). Here, microcavity was formed by inserting one defect layer into two DBRs. A schematic figure is provided in Figure 1B. The design of the SiO_2_ vertical microcavity and NPG parameters is crucial for developing a satisfactory plasmonic modes with higher detection sensitivities [23]. The effect of the top DBR bilayer numbers (N_T_), bottom DBR bilayer numbers (N_B_), NPG thickness (T_NPG_) and NPG porosity on resonance mode was studied through simulation. Meanwhile, to evaluate the sensitivity of different biosensors to RI change, we calculated the average shift relative to PBS buffer (pH = 7.4) under different RI. The setting of simulation parameters is provided in the Table 1 and the model of our biosensor is shown in Figure 1B.

### 2.3. Preclinical Validation of the Microcavity-LSPR Biosensor

#### 2.3.1. Preparation of Artificial Spiked SARS-CoV-2 Pseudovirus

SARS-CoV-2 spike pseudovirus was immersed in artificial saliva to build pseudo-SARS-CoV-2 saliva samples with viral burden from 10^0^ to 10^6^ copies/mL. MERS-CoV spike protein was diluted using PBS (pH = 7.4) to 1 μg/mL, 0.1 μg/mL, and 0.01 μg/mL.

#### 2.3.2. Equipment Set Up

The setup of the detection system is shown in Figure 2A. The real-time spectrum monitoring graphic user interface, the light source, the spectrometer, the Y-shaped optical fiber, the robotic arm system and the automatic detection software (The robot arm control software; LabVIEW code for robot arm-spectrometer communication and spectrum detection and recording; MATLAB code for data processing) were introduced in our previous work [6]. Here, the Y-shaped optical fiber was fixed on a robotic arm for automatic spectrum measuring and the axis of optical fiber (incidence) was perpendicular to the sensor surface (Figure 2B). A plate with a 10 × 10 pillar array (“plate”) was arranged under the optical fiber to fix the position of biosensor dies as shown in Figure 2C. The plate was designed according to the dimensions of standard 96-well plates to standardize the operation. Under this condition, the position of the biosensor die array was determined in different experiments.

The working principle of the automatic detection platform developed in house is illustrated in Figure 2D. Herein, the light source generates white incidence, and the reflected light of the biosensor die is received by the spectrometer. The robot arm is driven by the control software, and the robot arm–spectrometer communication is realized by the LabVIEW graphical user interface (GUI). This LabVIEW GUI is also responsible for real-time optical signal recording. During the detection, the robot arm drives the Y-shaped optical fiber to detect every biosensor dies on the plate in sequence. It takes approximately 100 s to record the reflectance spectrums of all the 100 biosensors. Meanwhile, the real-time reflectance spectrum can be viewed in the spectrometer software.

#### 2.3.3. Detection Processes of the Pre-Clinical Experiments

The spectrum was measured under wet conditions and the antigen–antibody binding reaction was performed in the surface solution. The detection involved an eight-step process: biosensor array construction; buffer loading; spectrum recording before sample loading; sample loading; biosensor washing; buffer loading; spectrum recording after sample loading.

In detail, first, biosensors with different porosities were stuck on the plates as shown in Figure 2C. We arranged eight biosensor dies for each concentration of SARS-CoV-2 pseudovirus and MERS-CoV spike protein sample detection. For each biosensor die, we added 20 μL of PBS buffer, covered with a glass slide on the sensor surface. This step ensured wet detection and guaranteed that the liquid surface was flat. Then, we used the robot arm system to record the reflectance spectrum of biosensors before loading the pseudovirus. The lower surface of optical fiber was 2 mm from the sensor surface. Then, we removed the residual PBS buffer and added SARS-CoV-2 pseudo-saliva samples with different viral loads to the biosensor dies. The SARS-CoV-2 pseudo-saliva sample solution was left on the chip surface for 15 min. Following this, we washed the biosensor dies with 50 μL of PBS buffer three times to remove the unbonded virus. Afterwards, 20 μL PBS buffer was added to each biosensor die, and a glass slide was used to cover its surface. Finally, after virus binding on the biosensor surface, the reflectance spectrum was recorded by the automatic detection system.

### 2.4. Statistics

We used the commercially available software Lumerical FDTD 2020 R 2.4 (ANSYS, Inc., DE, USA) for simulation. Origin 2018 (OriginLab Corporation, Northampton, MA, USA) was applied for data processing. The total shift (Equation (1)) is the sum of the absolute value of the six characteristic valleys’ difference in reflectance spectrum before and after sample binding. The six characteristic valleys contain one resonance valley (${\Delta \lambda }_{1}$) and five interference valleys (${\Delta \lambda }_{2}$ to ${\Delta \lambda }_{6}$) as shown in Figure 1C. In this study, the optical response signal is calculated by the average shift, which is the average value of total shift (Equation (2)).
(1)$$\Delta \lambda_{\mathrm{total}}=\sum_{i=1}^{6}|\Delta \lambda_{i}|,$$
(2)$$\Delta \lambda_{\mathrm{average}}=\bar{\Delta \lambda_{\mathrm{total}}},$$
where λ is the wavelength of the valley on the reflectance spectrum as shown in Figure 1C.

## 3. Results

### 3.1. Investigation of the NPG Parameters for the Microcavity-LSPR Biosensor

In our hybrid biosensor, NPG was formed to excite the LSPR. To study the effect of surface morphology on detection sensitivity, different porosities of NPG were deposited on the microcavity surface. Herein, the SiO_2_ vertical microcavity was formed by electrochemical anodization, with alternating current densities of 2.708 mA/cm^3^ (20 s) and 23.618 mA/cm^3^ (6 s). The defect layer was etched by a current density of 23.618 mA/cm^3^ for 6 s.

The choice of adhesive layer is a decisive factor in the successful formation of NPG thin film on SiO_2_ vertical microcavity. A satisfying adhesive layer should have good adhesion with the substrate and Au thin film layer in addition to guaranteeing biosensor stability. Herein, we tested the stability of Cr, Ni, and Ti as adhesive metals. Both Ni and Ti can suffer from peeling during the etching of Ag in an Ag–Au alloy, while Cr resulted in a stable NPG thin film layer on the microcavity substrate. Therefore, we first deposited 5 nm Cr as an adhesive layer, followed by 5 nm Au as a barrier layer to avoid the etching of the adhesive layer by HNO_3_. Afterwards, Ag and Au were simultaneously deposited on the biosensor die with five ratios, ranging from 0 to 1.5. Five different porosities (Au thin film, 10%, 20%, 40%, and 60% porosity NPG; thickness: 25 nm) were assessed. The deposition condition of each group is listed in Table 2.

### 3.2. Characterization of the Microcavity-LSPR Biosensor by SEM Imaging

The surface characterization of the microcavity surface formed with NPG and Au thin film was investigated by field emission scanning electron microscopy (SEM, Zeiss Gemini 500). SEM images of SiO_2_ microcavity, Au thin film and NPG deposited biosensors were shown in Figure 1C–F and Figure 3A,B, respectively. The Au thin film and NPG deposited biosensors are fabricated with the conditions summarized in Table 2. The NPG thin film resulted in larger effective binding surface for the antibody than that of the Au thin film, indicating the potential to achieve a higher detection sensitivity. The experimental NPG porosity was calculated by first increasing the contrast of the image by a factor of 5 and then gaining the percentage of pixels with gray scale between 0 and 0.4. For each porosity (10%, 20%, 40%, and 60%) of the biosensor die, we calculated the experimental porosities of ten samples as shown in Figure 4A–D. In summary, our NPG fabrication method can adjust the morphology of NPG thin film by controlling the PVD parameters. Satisfactory stability was realized for different morphologies of NPG thin film.

### 3.3. Investigation of the RI Sensing Performance via FDTD Simulation

After adopting the selected geometry model of NPG, we simulated the interactions between the microcavity performance with NPG parameters and microcavity structure through Lumerical FDTD. Through simulation, the microcavity and NPG parameters of N_T_ = 6, N_B_ = 10 and T_NPG_ = 25 nm was adopted to induce a better resonance mode. These parameters were used in the following work. The RI of PBS buffer (pH = 7.4) was set to be 1.32 and the sensitivity was analyzed by calculating the $\Delta \lambda_{\mathrm{average}}$ of the reflectance spectrum for each RI value relative to the reflectance spectrum of the PBS. Figure 5 shows the simulation result regarding the sensitivity of biosensors with different NPG thin film porosity to RI changes. Generally, surface morphology greatly affected biosensor sensitivity for monitoring RI change. The pure microcavity and Au thin film-deposited sensors were much less sensitive in all the RI range compared with NPG deposited biosensor dies, as reflected by the much lower average shifts of the two cases in Figure 5. However, the sensitivity of the NPG-deposited sensors differed in the RI ranges. Generally, 40%-porosity NPG thin film had the highest sensitivity in the studied RI range, followed by 60%, 20%, and 10% porosity NPG-deposited biosensors. This demonstrates the dependence of resonance sensitivity on NPG porosity. Higher porosity allows NPG to better interact with the surrounding medium due to the larger overlap between the surface plasmon mode and the analyte. Thus, with the increase of porosity to 40%, our structure showed better sensitivity. However, when the porosity reached 60%, the large metal gap weakened the LSPR and light–matter interaction, decreasing sensitivity.

### 3.4. Pre-Clinical Validation of the Microcavity-LSPR Biosensor Using the Artificial Spiked SARS-CoV-2 Pseudovirus Saliva Samples

The five porosity NPG-deposited biosensors 10%, 20%, 40%, and 60% were arranged to detect SARS-CoV-2 pseudovirus with viral loads ranging from 10^0^ to 10^6^ copies/mL. We also detected the response to artificial saliva buffer (marked as “Blank”) and MERS-CoV spike protein (marked as “N1”,” N2”, and “N3” for 1 μg/mL, 0.1 μg/mL and 0.01 $\frac{\mu g}{\mathrm{mL}},$ respectively). In samples containing the target virus, the antigen–antibody binding will lead to a significant shift of the six characteristic valleys in the reflectance spectrum, whereas for samples containing no target analyte, negligible signal shifts are expected.

The optical response signal of each experiment group is shown in Figure 6. For each concentration of pseudovirus, we arranged eight biosensor dies with different porosities for the experiment. The error bars in Figure 6A–E represent the 95% confidence interval (CI) of the optical response signal of eight chips for each viral load. The wavelength of the red dash-dot line (named “${\Delta \lambda }_{\mathrm{ref}}$”) in Figure 6A–E can be calculated by Equation (3):(3)$${\Delta \lambda }_{\mathrm{ref}}={\Delta \lambda }_{\mathrm{average}-\mathrm{blank}}+3\sigma_{\mathrm{blank}},$$
where ${\Delta \lambda }_{\mathrm{total}-\mathrm{blank}}$ and $\sigma_{\mathrm{blank}}$ is the average shift and the standard deviation of the blank group, respectively. The LoD can be calculated by Equation (4):(4)$$\mathrm{LoD}=\frac{\left({\Delta \lambda }_{\mathrm{ref}}-{\Delta \lambda }_{n-1}\right)\;\times \;\left({\mathrm{VL}}_{n}-{\mathrm{VL}}_{n-1}\right)}{\left({\Delta \lambda }_{n}-{\Delta \lambda }_{n-1}\right)}+{\mathrm{VL}}_{n-1},$$
where ${\mathrm{VL}}_{n}$ and ${\mathrm{VL}}_{n-1}$ are the upper and lower adjacent viral loads of the ${\Delta \lambda }_{\mathrm{ref}}$, respectively, and ${\Delta \lambda }_{n}$ and ${\Delta \lambda }_{n-1}$ are the average shift of upper and lower adjacent viral loads of the ${\Delta \lambda }_{\mathrm{ref}}$, respectively.

The experimental results revealed a decrease–increase relationship between LoD and the change of porosity (Au thin film is considered as 0 porosity here). The 40% porosity NPG-deposited biosensor was more sensitive to the antibody-virus binding than other biosensors, resulting in an LoD of 319 copies/mL. This result is consistent with the simulation of RI sensing. Moreover, all biosensors had no cross-reaction with MERS-CoV spike protein at 1 μg/mL concentration, and the biosensing systems detected the 10 × 10 detection array (100 samples) within 30 min, which is suitable for rapid onsite SARS-CoV-2 diagnosis. Meanwhile, biosensor surface morphology can affect the sensing performances. According to the experiment results of MERS-CoV spike protein, blank, 10^0^, 10^1^ and 10^2^ copies/mL SARS-CoV-2 pseudovirus, the rough surface can trap a portion of unbonded antigen and residual buffer on the biosensor surface. This is reflected by the higher experimental signal values of NPG coated biosensors compared to Au thin film coated biosensor. Among these, the 40% porosity NPG deposited biosensor has the highest optical response signal, followed by 60% and 20% porosity NPG deposited biosensors. The 10% porosity NPG coated biosensor has the lowest low-viral load experimental response signal compared to the other NPG deposited biosensors. As such, we can conclude that for biosensor with no or small pores on surface, very few unbounded viruses will be trapped on the surface. In addition, when the pore size is large, the unbonded virus is easy to be removed by washing the biosensor surface. This further validates that we can improve the sensitivity of biosensor by controlling the surface morphology of biosensor.

Compared to other SARS-CoV-2 biosensors, the developed microcavity-LSPR biosensor showed multiple advantages, including a relatively shorter time to obtain results, good sensitivity, full automation, and high-throughput (Table 3).
Figure 6Detection performance of (**A**) Au thin film, (**B**) 10%, (**C**) 20%, (**D**) 40%, and (**E**) 60% porosity NPG deposited SiO_2_ vertical microcavity biosensors. (**F**) Variation of LoD with the NPG porosity.
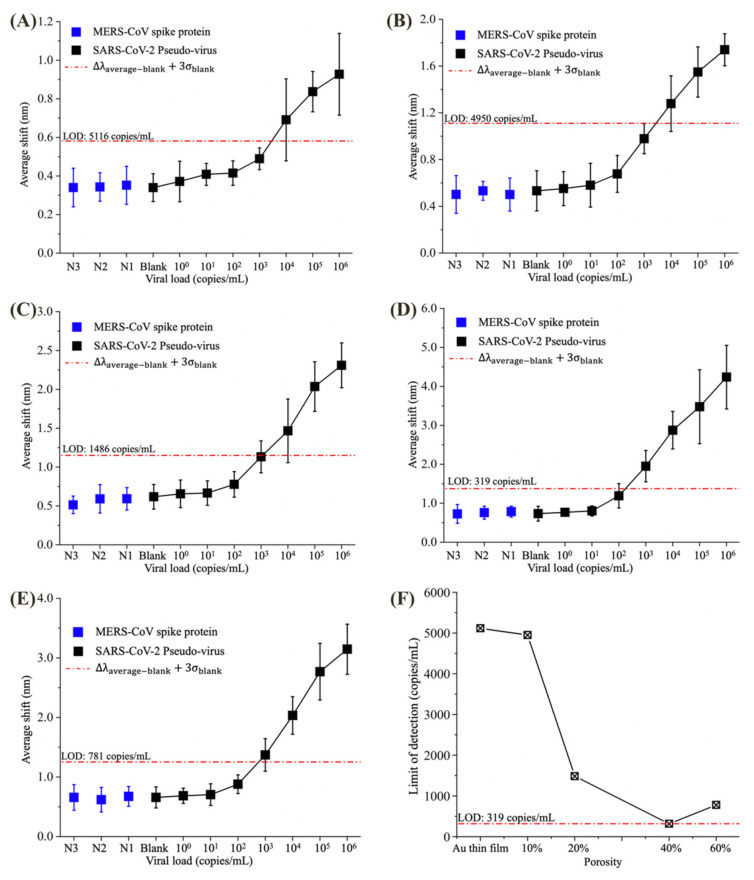


## 4. Discussion and Conclusions

This work adopted a low-cost and convenient technique to form an NPG thin film on SiO_2_ vertical microcavity and established a vertical microcavity-LSPR hybrid biosensor for SARS-CoV-2 detection with high sensitivity and throughput. Biosensors with different NPG thin film porosities were fabricated. FDTD simulation was carried out to improve the design of the biosensor structure and study the influence of NPG porosity on the RI sensing sensitivity. Afterwards, experiments using SARS-CoV-2 pseudovirus spiked in artificial saliva were performed to study the dependance of biosensor sensitivity on NPG porosity. The simulation showed that the design of NPG morphology greatly impacts RI detection sensitivity. The experimental results further revealed that a biosensor die with 40% NPG porosity (25 nm thickness) could achieve an LoD of 319 copies/mL. Moreover, the achieved biosensor had no cross reaction with MERS-CoV spike protein and could detect 100 samples within 30 min.

Collectively, the vertical microcavity-LSPR hybrid biosensing system holds multiple novel advantages, including the short turnaround time, full automation, and good sensitivity and in high-throughput, making it promising for on-site monitoring of SARS-CoV-2 in clinical practice. Compared to typical LSPR biosensors, the hybrid biosensor has a compact structure and requires only a limited sample of 20 μL for detection. Compared to a single vertical microcavity sensor, the hybrid biosensor has much higher sensitivity. However, our developed hybrid biosensor is still in the early stages of development. This proof-of-concept technique requires large-scale real samples to systematically validate its clinical utility. Meanwhile, the detection performance remains to be improved, such as the incubation time of the samples. In addition, this biosensing system will be applied for other types of human coronavirus detection, to deal with the related periodic epidemics.

## Figures and Tables

**Figure 1 biosensors-12-00151-f001:**
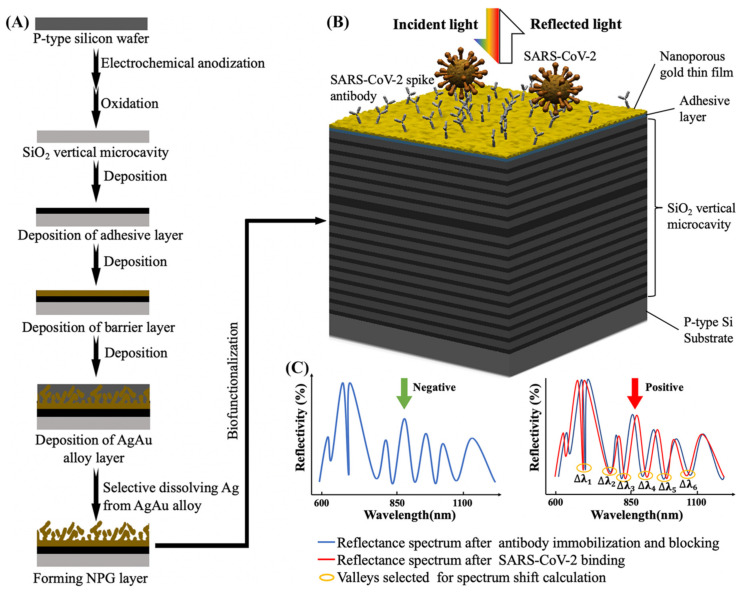
Diagram and detection mechanism of the LSPR- SiO_2_ vertical microcavity biosensor: (**A**) the process flow in biosensor fabrication, including the p-type silicon wafer, the SiO_2_ vertical microcavity, the adhesive layer deposition, the barrier layer deposition, the Ag-Au alloy thin film deposition and the NPG deposited biosensor die. (**B**) Schematic illustration of the structure of microcavity-LSPR biosensor. The sensor surface is immobilized with SARS-CoV-2 neutralizing antibodies for SARS-CoV-2 capture. (**C**) The optical signal response detection. If the sample contains target virus, a shift in reflectance spectrum will be observed; if no target virus presents, zero shift in the spectrum shift will be observed.

**Figure 2 biosensors-12-00151-f002:**
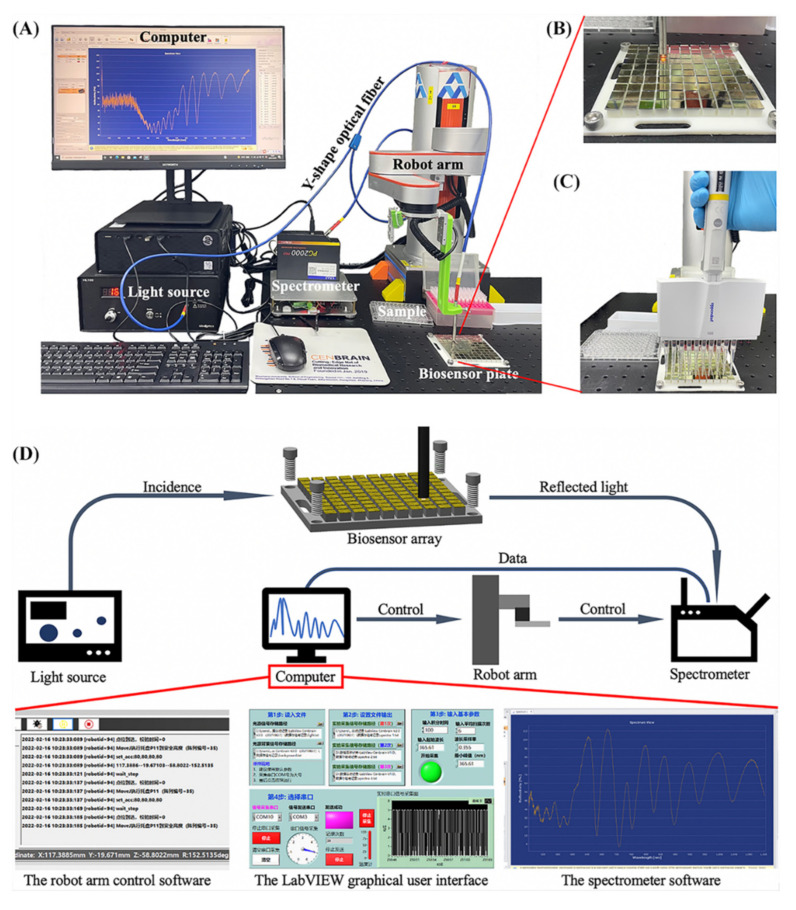
The automatic detection system used in this study. (**A**) The robot arm system for automatic detection including the computer, the light source, the spectrometer, the Y-shaped optical fiber, the sample, and the biosensor detection array; (**B**) zoom in of the automatic measurement of the reflectance spectrum; (**C**) the automated sampling process empowered by multi-channels; (**D**) the working principle of the automatic detection system.

**Figure 3 biosensors-12-00151-f003:**
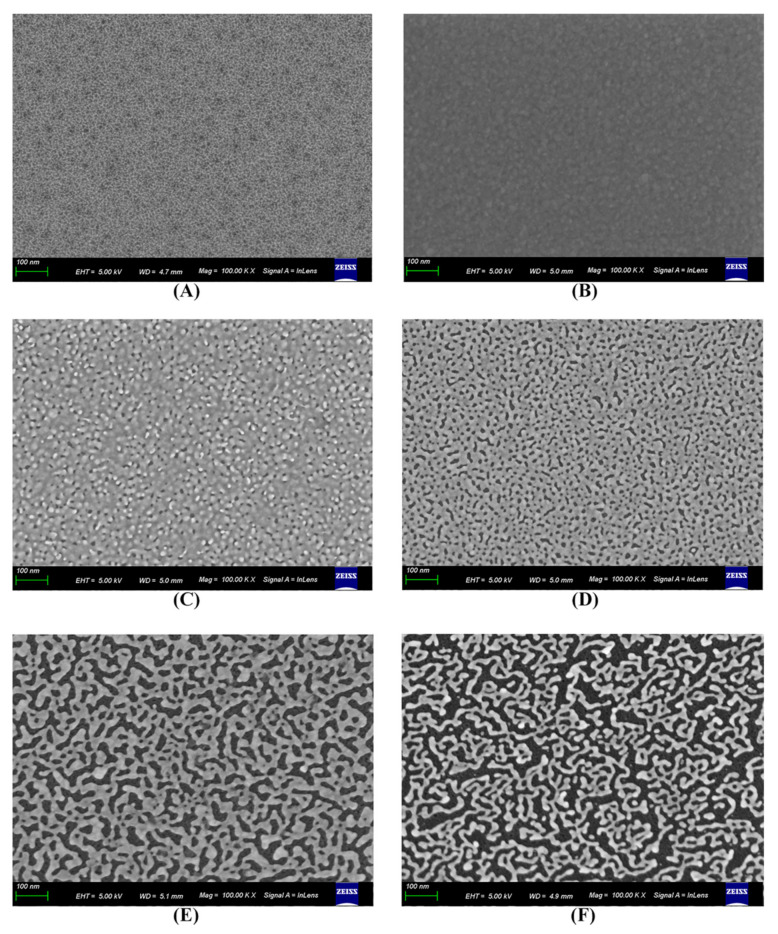
Surface SEM images of: (**A**) SiO_2_ vertical microcavity wafer. (**B**) Au thin film deposited SiO_2_ vertical microcavity biosensor die; (**C**) 10%, (**D**) 20%, (**E**) 40%, and (**F**) 60% porosity NPG deposited SiO_2_ vertical microcavity biosensor die.

**Figure 4 biosensors-12-00151-f004:**
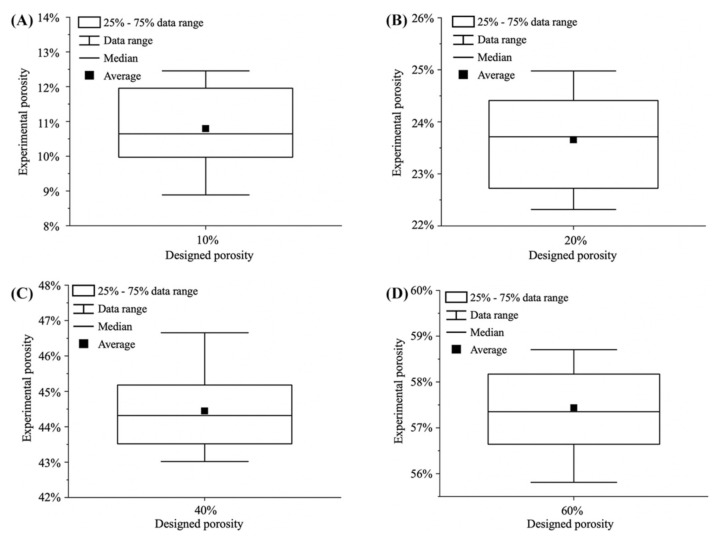
Designed porosity and experimental porosity of (**A**) 10%, (**B**) 20%, (**C**) 40%, and (**D**) 60% porosity NPG deposited biosensors. The error bar in the figure represents the data range.

**Figure 5 biosensors-12-00151-f005:**
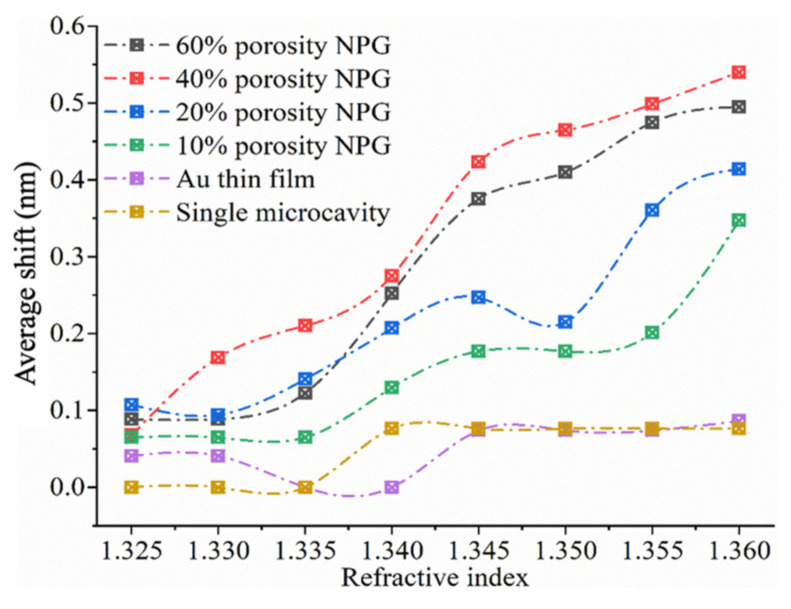
Simulation results of RI-sensing performance of microcavity-LSPR hybrid biosensors with different NPG porosities.

**Table 1 biosensors-12-00151-t001:** Simulation parameters.

Layer	Material	RI	Geometry Parameters	Mesh Setting
NPG-Au thin film	Au (CRC)	RI of Au (CRC)	25 nm	dx: 5 nm dy: 5 nm dz: 1 nm
Au thin film	Au (CRC)	RI of Au (CRC)	5 nm	
Adhesive layer	Cr (CRC)	RI of Cr (CRC)	5 nm	
Top DBR	SiO2	1.65/1.22	100 nm/135 nm	
Cavity	SiO2	1.22	135 nm	
Bottom DBR	SiO2	1.65/1.22	100 nm/135 nm	
Substrate	Si (Palik)	RI of Si (Palik)	2 mm	

DBR: distributed Bragg reflector; NPG: nanoporous gold; RI: refractive index; SiO_2_: Silica; T_Au_: thickness of Au thin film layer.

**Table 2 biosensors-12-00151-t002:** Deposition conditions in forming NPG thin film with different porosities on SiO_2_ vertical microcavity wafer.

Group	Designed Porosity	Rate of Deposition (A/s)		Deposition Time (s)
		Ag	Au	
1	0	0	0.50	500
2	10%	0.05	0.45	500
3	20%	0.10	0.40	500
4	40%	0.20	0.30	500
5	60%	0.30	0.20	500

**Table 3 biosensors-12-00151-t003:** Performance of the latest biosensors for SARS-CoV-2 detection reported in literature.

Biosensor Type	Target Analyte	Sample	LoD (Copies/mL)	Detection Time	Number of Samples Per Test	Operation	Ref.
Electrochemical	ORF1ab	Throat/oral swabs, sputum, urine, plasma, feces	200	~3 h	Single	By hand	[3]
Electrochemical	N gene	Nasopharyngeal/ oropharyngeal swabs	2.58 × 10^5^	15 min	Single	By hand	[4]
Electrochemical	N and S genes	Nasopharyngeal swab	1000	<2 h	Single	By hand	[5]
Graphene-based FET	S protein	Nasopharyngeal swab	242	~120 s	Single	By hand	[7]
LSPR-microcavity	Pseudovirus	Artificial saliva	319	<30 min	Up to 100 samples	Automated	This work

FET: field-effect transistor; ORF1ab: open reading frame 1ab; LoD: limit of detection; Ref.: Reference.

## Data Availability

Not applicable.

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
