# Peer review of "Label-Free LSPR-Vertical Microcavity Biosensor for On-Site SARS-CoV-2 Detection"

_biosensors, 2022, doi:10.3390/bios12030151_

Round 1

Reviewer 1 Report

The manuscript submitted by Prof. Mohamad Sawan and coworkers reported e a vertical microcavity and localized surface plasmon resonance hybrid biosensor for SARS-CoV-2 detection in artificial saliva and assessed its efficacy. The proposed biosensor monitors the valley shifts in the reflectance spectrum, as induced by changes in the refractive index within the proximity of the sensor surface. Some issues should be solved before publication. The work would be more interesting if deep study was added. The work can be published after the following issues are addressed.

(1) As the proposed sensing strategy is a vertical microcavity-LSPR hybrid biosensor for on-site

SARS-CoV-2 detection, it is very important for authors to discuss the progress of LSPR sensors for the detection and then point out the novelty or distinctive feature of the used nanoporous gold in the introduction part, which may deeply highlight the motivation of this work.

(2) the authors can adjust the morphology of NPG thin film by controlling the PVD parameters, so It is necessary for authors to discuss the relationship between the morphology and sensing performances.

(3) it seems that some SEM images in figure 3 are not clear, the authors should try to improve them.

(4) The authors showed the automatic detection system used in this study, but it seems all images from photography, the authors should give a schematic process about the mechanism for the sensing.

(5) the authors should do more experiments about specificity and reproducibility of the proposed sensing strategy.

Author Response

Dear reviewer,   We appreciate every suggestion to improve our paper. We did our best to address your comments. Please see the attachment.   Thank you very much.

Reviewer 2 Report

This manuscript has described a silicon wafer-based nanoporous gold sensor for SARS-Cov-2 detection. The limit of detection was extraordinary (327 copies/mL) in artificial saliva. Here I have some questions.

  1. The experiment takes 30 mins. I am wondering whether the author has done any incubation time optimization. Currently, 30 mins is relatively long compared with other techniques.
  2. I am curious about the antibody density on the surface. Does the author have any experiments to quantify the antibody density? Generally, people use AFM mapping, but it may not suit the rough surface.
  3. In figure 1B, the author shows a stacking array, but in Figure 2B, the signal was read out one by one. Please clarify this.
  4. The author was using an inactive virus spiked in artificial saliva. I am wondering if it is possible to do a real sample. Also, the author needs to do RT-qPCR to quantify the virus copy in the artificial saliva. 
  5. SEM image quality can be improved. I am not sure what detector was used. I would also suggest trying the T1 or T2 in lens detector in UHR (ultra-high resolution) mode or immersion mode at a WD 5mm, 5kV, spot size 1-4, with a detector bias set to +280 SE/BSE, immersion mode. 
  6. The surface is quite rough, may the antibody or the virus trap in the fine structure? The binding relies on thermal diffusion. However, the fine surface structure is going opposite. Will this affect the sensitivity or detection limit? It is necessary to discuss.

Author Response

(The authors gave the same response as above.)

Round 2

Reviewer 2 Report

This version looks much better than before. It is ready to be published.